# A Primal-Dual Approach for Dynamic Pricing of Sequentially Displayed Complementary Items under Sale Constraints

## Abstract

We address the challenging problem of dynamically pricing complementary items that are sequentially displayed to customers. An illustrative example is the online sale of flight tickets, where customers navigate through multiple web pages. Initially, they view the ticket cost, followed by ancillary expenses such as insurance and additional luggage fees. Coherent pricing policies for complementary items are essential because optimizing the pricing of each item individually is ineffective. Our scenario also involves a sales constraint, which specifies a minimum number of items to sell, and uncertainty regarding customer demand curves. To tackle this problem, we originally formulate it as a Markov decision process with constraints. Leveraging online learning tools, we design a primal-dual online optimization algorithm. We empirically evaluate our approach using synthetic settings randomly generated from real-world data, covering various configurations from stationary to non-stationary, and compare its performance in terms of constraints violation and regret against well-known baselines optimizing each state singularly.

## 1 Introduction

Dynamic Pricing (DP) aims to determine the ideal pricing for a product or service in real-time employing revenue optimization strategies (Rothschild, 1974; Kleinberg & Leighton, 2003; Trovò et al., 2018). This practice is widely prevalent in various sectors, including airlines, ride-sharing, and retail, owing to its capability to adapt to variables such as demand, competition, and time constraints. Undoubtedly, dynamic pricing is garnering considerable attention from both the industry and the scientific community due to its profound economic impact on businesses. From a scientific standpoint, while early research in this field assumed knowledge of the underlying demand functions, the imperative for real-world AI applications has prompted the scientific community to shift their focus towards uncharted demand scenarios and exploration-exploitation algorithms, as underscored by seminal works (Aviv & Pazgal, 2005; Besbes & Zeevi, 2009)). Moreover, research on non-stationary demand functions has made a substantial impact on the field. Specifically, recent studies have concentrated on external non-stationarity factors, such as internal ones driven by the seller's actions (Cui et al., 2023), as well as seasonality (Besbes & Saure, 2014). From an industrial standpoint, the proliferation of e-commerce platforms and online marketplaces has led to a significant acceleration in data collection and availability, thereby fostering the development of data-driven dynamic pricing algorithms. It is not an exaggeration to regard these two categories of retailers as the primary beneficiaries of dynamic pricing strategies.

The use of reinforcement learning (RL) (Sutton & Barto, 2018) and Markov decision processes (MDP) in dynamic pricing contexts is widely recognized. RL methodologies, such as Q-learning, are extensively employed to address discrete pricing issues within the MDP framework. This is largely due to their effectiveness in navigating the uncertainties associated with demand and pricing results (e.g., (Kim et al., 2015; Lu et al., 2018; Liu et al., 2021)). The same reasoning holds for simpler single-state dynamic pricing settings, where Multi-Armed Bandits algorithms (Auer et al., 2002; Cesa-Bianchi & Lugosi, 2006) and online learning techniques (Hazan, 2019; Orabona, 2019) are used to learn the optimal price, while keeping small the losses that the seller may incur during

the learning dynamic (see (Ganti et al., 2018; Misra et al., 2019; Trovò et al., 2018)). Due to space constraints, we refer to Appendix A for a complete discussion on related works.

Our study centers on the pricing of complementary items that are sequentially presented to customers. To illustrate, consider the online sale of flight tickets, where customers navigate through multiple web pages. Initially, they view the base ticket cost, followed by ancillary expenses such as insurance and additional luggage fees. Coherent pricing policies play a pivotal role in optimizing the overall value of these complementary items, as individually optimizing the price of each item may lead to arbitrarily inefficient solutions. We delve into two additional features commonly encountered in real-world applications. First, we address the uncertainty surrounding customer demand curves, emphasizing the need for online machine learning tools. In particular, we explore both stationary and non-stationary settings. Second, we examine constraints related to pricing optimization. Specifically, we impose a requirement that the number of sales should not fall below a certain threshold. Interestingly, managing these constraints when learning online poses significant technical challenges that are still open in the scientific literature.

We first formulate the problem by adopting the Markov decision process (MDP) framework with constraints (CMDP) (Altman, 1999), where the sale constraint is encoded within the violation definition. In each state of the CMDP, the seller must choose the price for an item, subsequently observing the associated reward and any incurred violations. To the best of our knowledge, this is the first work to employ the online CMDP mathematical framework to solve dynamic pricing scenarios. Thus, to address the problem, we introduce a primal-dual online learning algorithm that optimizes the seller's expected reward while minimizing the number of violations. Notably, our algorithm is tailored to handle non-stationary demand curves, eliminating the need for stationarity assumptions regarding reward and constraint functions. Finally, we empirically evaluate the performance of our algorithm. To do so, we generate an online simulator from real-world data from one of the world's leading online travel agencies. This empirical evaluation demonstrates the effectiveness of our approach in maximizing the expected rewards while minimizing the violations of constraints.

## 2 PROBLEM FORMULATION

In our study, we examine a scenario where an e-commerce platform aims to sell a primary product $i_1$ and a related ancillary item $i_2$ (for instance, a flight ticket and corresponding baggage). It is crucial to price these items by considering their interdependence. Indeed, the pricing of the primary product (and similarly, the ancillary item) can influence the sales of the other. This is because potential customers might base their decision to purchase either or both items on the price of just one. For instance, if a customer needs to carry an extra bag on a flight and finds the baggage fee excessively high, they might buy his tickets from a competitor. Therefore, it's imperative to strike a balanced trade-off between the two prices to maximize total revenue. Additionally, the platform has a requirement that a specific portion of the revenue and a certain number of sold tickets are ensured by the set prices.

We formalize the aforementioned scenario employing the following mathematical model, known in the literature as an online Constrained Markov Decision Process (CMDP, (Altman, 1999)).

**Remark 1.** *Please notice that, even if the paper focuses on the pricing of two products (the main one and the related ancillary), both the model and the proposed algorithm can be easily extended to take into account more than two items.*

### 2.1 MATHEMATICAL FORMULATION OF THE MODEL

We model the *dynamic pricing of complementary items under sale constraint* setting through the mathematical framework defined by the tuple $M = \left( X, A, P, \{r_t\}_{t=1}^{T}, \{g_t\}_{t=1}^{T} \right)$, where:

- $T$ is an upper bound to the number of customers that visit the website, namely, the number of episodes of the online CMDP, with $t \in [T]$ denoting a specific episode.
- $X$ is the set of states. Precisely $X = \{x_0, x_1, x_2, x_3, x_4, x_5, x_6\}$, where $x_0$ is the primary item state, that is, the price of $i_1$ shown to the client. The states $x_1$ and $x_2$ represent ancillary states within our model. State $x_1$ is entered when the user decides to purchase at least the main product, while state $x_2$ is reached when the user has engaged with the

main product but chosen not to buy it, yet remains on the platform (website). Thus, both $x_1$ and $x_2$ refer to the page where $i_2$ is shown to the client. Please note that the uncertainty regarding the client's preferences leads to partial observability between states $x_1$ and $x_2$. Indeed, during the inference phase, the website is unable to differentiate between these states because the purchase occurs at the process's conclusion. However, this issue does not arise when the website updates its belief of the (unknown) environment. This update occurs after the complete traversal of the MDP, at which point the entire MDP has been revealed to the website. $x_4$ payment page state, where the user can eventually conclude the transaction or leave the website. Finally, $x_3, x_5, x_6$ are exit states. For the sake of simplicity, $X$ is partitioned into $L$ layers $X_0, \ldots, X_{L-1}$ with $L = 4$, such that the first and the last layers are singletons, *i.e.*, $X_0 = \{x_0\}$ and $X_3 = \{x_6\}$. Moreover, we have $X_1 = \{x_1, x_2, x_3\}$ and $X_2 = \{x_4, x_5\}$ (for more details see Figure 1).

- $A = \{A_0, A_1\}$ is the finite set of actions. Actions are only available in states $x_0, x_1, x_2$. The actions belonging to state $x_0$ (namely, $a \in A_0$) are the discretized prices of $i_1$, while the actions in state $x_1$ and $x_2$ (namely, $a \in A_1$) are the discretized prices of $i_2$.

- $P : X \times A \times X \to [0, 1]$ is the transition function, where, for simplicity in notation, we denote by $P(x'|x, a)$ the probability of going from state $x \in X$ to $x' \in X$ by taking action $a \in A$. By the loop-free property, it holds that $P(x'|x, a) > 0$ only if $x' \in X_{k+1}$ and $x \in X_k$ for some $k \in [0 \ldots L - 1]$. Transitions from state $x_0$ represent the probability of a user clicking on the primary product and either progressing to states $x_1$ or $x_2$ or opting to leave the website, which corresponds to transitioning to state $x_3$. Similarly, transitions from state $x_2$ depict the likelihood of the user engaging with the ancillary product, advancing to the final payment state $x_4$ (but without generating the final conversion), or deciding to exit the website, leading to the transition to the exit state $x_5$.

- $\{r_t\}_{t=1}^T$ is a sequence of vectors describing the rewards at each episode $t \in [T]$, namely $r_t \in [0, 1]^{|X \times A|}$. We refer to the reward of a specific state-action pair $x \in X, a \in A$ for a specific episode $t \in [T]$ as $r_t(x, a)$. Rewards may be *stochastic*, in that case, $r_t$ is a random variable distributed according to a fixed probability distribution $\mathcal{R}$ for every $t \in [T]$, but also highly non-stationary, namely $r_t \sim \mathcal{R}_t$ where $\mathcal{R}_t$ is the rewards distribution at episode $t \in [T]$. Reward at exit states $x_3, x_5$ and $x_6$ is always 0. Reward $r(x_0, a)$ measures the profit from selling the main product at price $a$. Similarly, the reward at state $x_1$ (namely, $r(x_1, a)$) is the profit from selling the ancillary product, plus a constant bonus for clicking on the main product. The same constant bonus is applied in $r(x_2, \cdot)$ and $r(x_4, \cdot)$ for engagement with the primary and ancillary product, respectively. Note that rewards affect transitions:
  1. If a user buys the primary product in state $x_0$, i.e., the reward is not equal to zero, they move to state $x_1$. Here, regardless of whether they purchase an ancillary product or not, they will eventually reach the final payment state.
  2. Conversely, a zero reward in state $x_0$ implies no primary product purchase. This leads to two possibilities: the user either exits or moves to state $x_2$. In state $x_2$, since the primary product was not purchased, the user can't buy the ancillary product but receives the predefined bonus. The user will then either exit or go to the payment state without any purchase.

- $\{g_t\}_{t=1}^T$ is a sequence of constraint functions describing the *constraint* violations at each episode $t \in [T]$, namely $g_t \in [-1, 1]^{|X \times A|}$, where non-positive violation values stand for satisfaction of the constraints. We refer to the violation of the $i$-th constraint for a specific state-action pair $x \in X, a \in A$ at episode $t \in [T]$ as $g_t(x, a)$. Constraint violations could be *stochastic* (in that scenario, $g_t$ is a random variable distributed according to a probability distribution $\mathcal{G}$ for every $t \in [T]$) or highly non-stationary, namely, $g_t \sim \mathcal{G}_t$. The objective of the constraints is to maintain a minimum ratio of product sales to website views, thus preventing overly aggressive pricing tactics. For states other than $x_0$, the constraint value is set to zero. In $x_0$, the constraint function is expressed as $g_t(x_0, a) = \tau - \mathbb{I}_{\{r_t(x_0, a) \neq 0\}}$, where $\tau$ is the target ratio of sales to views, and the second term is the indicator function on the sale of the primary product. The constraint value is positive if the primary product is not purchased and negative otherwise. Therefore, the constraint is only met if the average proportion of sales exceeds $\tau$.

Figure 1: The (constrained) Markov decision process employed to model the dynamic pricing of complementary items problem. The states colored in blue are the ones where the agent (website) chooses the prices. The orange ones are the exit states, while the grey one is the payment page state.

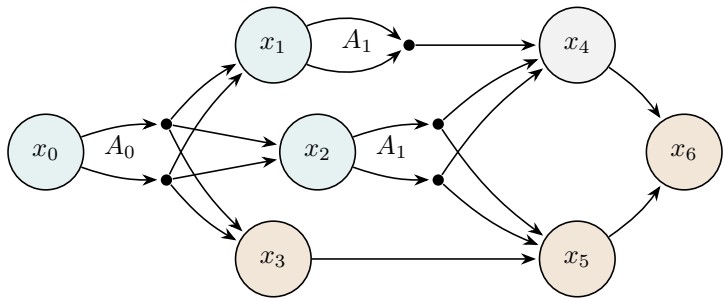

The website chooses a *pricing policy* $\pi : X \times A \to [0, 1]$ at each episode, defining a probability distribution over actions (prices) at each state. For simplicity in notation, we denote by $\pi(\cdot|x)$ the probability distribution for a state $x \in X$, with $\pi(a|x)$ denoting the probability of action $a \in A$.

In Algorithm 1, we report the interaction between customers and the website in the online CMDP formulation.

## 2.2 MATHEMATICAL TOOLS

To enhance understanding, it is essential to introduce the concept of the *occupancy measure*, as detailed in (Rosenberg & Mansour, 2019a). Informally, the occupancy measure represents, for each possible triple $(x, a, x')$, the probability that the triple is visited by the customer during the permanence on the website. Formally, given a transition function $P$ and a policy $\pi$, the occupancy measure

---

**Algorithm 1** Customer-Website Interaction

1: **for** $t = 1, \ldots, T$ **do**
2:     A potential customer enters the website
3:     The website chooses a pricing policy $\pi_t : X \times A \to [0, 1]$
4:     **for** $k = 0, \ldots, L - 1$ **do**
5:         The website chooses price $a_k \sim \pi_t(\cdot|x_k)$
6:         The website evolves to page $x_{k+1} \sim P(\cdot|x_k, a_k)$ given the customer choice, without distinguishing $x_1, x_2$
7:     The website observes $r_t(x_k, a_k)$ and $g_t(x_k, a_k)$ given the customer behaviour, for any $(x_k, a_k)$ traversed

---

$\boldsymbol{q}^{P,\pi} \in [0, 1]^{|X \times A \times X|}$ induced by $P$ and $\pi$ is such that, for every $x \in X_k$, $a \in A$, and $x' \in X_{k+1}$ with $k \in [0 \ldots L - 1]$ $q^{P,\pi}(x, a, x') = \Pr[x_k = x, a_k = a, x_{k+1} = x'|P, \pi]$. Moreover, we define $q^{P,\pi}(x, a) = \sum_{x' \in X_{k+1}} q^{P,\pi}(x, a, x')$ and $q^{P,\pi}(x) = \sum_{a \in A} q^{P,\pi}(x, a)$. Following (Rosenberg & Mansour, 2019b), for every $\boldsymbol{q} \in [0, 1]^{|X \times A \times X|}$, it holds that $q$ is a valid occupancy measure of an episodic loop-free MDP if and only if, the following three conditions hold: (i) $\sum_{x \in X_k} \sum_{a \in A} \sum_{x' \in X_{k+1}} q(x, a, x') = 1, \forall k \in [0 \ldots L - 1]$, (ii) $\sum_{a \in A} \sum_{x' \in X_{k+1}} q(x, a, x') = \sum_{x' \in X_{k-1}} \sum_{a \in A} q(x', a, x), \forall k \in [1 \ldots L - 1], \forall x \in X_k$ and (iii) $P^q = P$, where $P$ is the transition function of the MDP and $P^q$ is the one induced by $q$ (see Equation (1)).

Notice that any valid occupancy measure $q$ induces a transition function $P^q$ and a policy $\pi^q$ as:

$$P^q(x'|x, a) = \frac{q(x, a, x')}{q(x, a)}, \quad \pi^q(a|x) = \frac{q(x, a)}{q(x)}. \tag{1}$$

In the following, we will refer as $\boldsymbol{q}_t$ to the occupancy induced by policy $\pi_t$ chosen by the learner. Furthermore, given the definition of occupancy measure, it is possible to define the objectives of the learner. Precisely, the learner aims to maximize the rewards accumulated during the episodes, in expectation over policies and transitions, namely $\sum_{t=1}^{T} \boldsymbol{r}_t^\top \boldsymbol{q}_t$. Similarly, regarding the fulfillment of constraints, the goal is to minimize the occurrences of constraint violations $\sum_{t=1}^{T} \boldsymbol{g}_t^\top \boldsymbol{q}_t$.

---

**Algorithm 2** Primal-Dual Bandit Online Policy Search for Dynamic Pricing (PD-DP)

---

**Require:** State space $X$, action space $A$, episode number $T$, learning rate $\eta$, confidence parameter $\delta$, parameter $\alpha$

1: Initialize epoch index $i = 1$, confidence set $\mathcal{P}_1$ as set of all transition functions. Initialize counters for all $k = 0, \ldots, L-1$ and all $(x, a, x') \in X_k \times A \times X_{k+1}$ to 0, i.e., $N_i(x, a) = M_i(x'|x, a) = 0 \, \forall (x, a, x') \in X_k \times A \times X_{k+1}, \, \forall k = 0, \ldots, L-1, \forall i = 0, 1$. Initialize occupancy measure, policy, and Lagrangian variable

$$\widehat{q}_1(x, a, x') = \frac{1}{|X_k||A||X_{k+1}|} \; ; \; \pi_1 = \pi^{\widehat{q}_1} \; ; \; \lambda_1 = 0$$

2: **for** $t = 1$ to $T$ **do**
3:  Play policy $\pi_t$ and observe $x_k, a_k, r_t(x_k, a_k), g_t(x_k, a_k)$ for $k = 0, \ldots, L-1$
4:  Build loss as $\ell_t(x_k, a_k) = g_t(x_k, a_k) \cdot \lambda_t - r_t(x_k, a_k)$
5:  Compute upper occupancy bound $u_t(x_k, a_k) = \text{Comp-UOB}(\pi_t, x_k, a_k, \mathcal{P}_i)$ for each $k = 0, \ldots, L-1$
6:  Build loss estimators for all $(x, a)$:

$$\widehat{\ell}_t(x, a) = \frac{\ell_t(x, a)}{u_t(x, a) + \eta} \mathbb{I}_t(x_{k(x)} = x, a_{k(x)} = a)$$

7:  Update counters $N_i(x_k, a_k) \leftarrow N_i(x_k, a_k) + 1$, and $M_i(x_k, a_k, x_{k+1}) \leftarrow M_i(x_k, a_k, x_{k+1}) + 1$,
8:  **if** $\exists k, N_i(x_k, a_k) \geq \max\{1, 2N_{i-1}(x_k, a_k)\}$ **then**
9:   Increase epoch index $i$
10:   Initialize new counters $N_i(x_k, a_k) = N_{i-1}(x_k, a_k)$, $M_i(x_k, a_k, x_{k+1}) = M_{i-1}(x_k, a_k, x_{k+1})$
11:   Update confidence set $\mathcal{P}_i$ based on Equation (3)
12:  Update occupancy measure $\widehat{q}_{t+1} = \arg\min_{q \in \Delta(\mathcal{P}_i)} \widehat{\ell}_t^\top q + \frac{1}{\eta} D(q \parallel \widehat{q}_t)$
13:  Update dual variable as $\lambda_{t+1} \leftarrow \Pi_{\mathbb{R}_{\geq 0}} \left( \lambda_t + \eta \sum_k g_t(x_k, a_k) \widehat{q}_t(x_k, a_k) \right)$
14:  Compute policy $\pi_t \leftarrow \widehat{q}_t$, with $\pi_t(x_1, \cdot) = \pi_t(x_2, \cdot) := \alpha \pi_t(x_1, \cdot) + (1 - \alpha) \pi_t(x_2, \cdot)$

---

## 3 ALGORITHM

In line with the typical structure of primal-dual algorithms, our method involves implementing two separate optimization processes. From a game-theoretic standpoint, these are identified as the primal and dual players in the Lagrangian game, which is defined as follows:

$$\max_{\boldsymbol{q} \in \Delta(M)} \min_{\lambda \in \mathbb{R}_{\geq 0}} \boldsymbol{r}^\top \boldsymbol{q} - \lambda(\boldsymbol{g}^\top \boldsymbol{q}), \tag{2}$$

for any reward $\boldsymbol{r}$ and constraint $\boldsymbol{g}$ vectors. This concept aligns with the framework described in (Stradi et al., 2024b) for adversarial CMDPs. Formally, the primal player focuses on optimizing within the primal variable space of the Lagrangian game, specifically, the set of valid occupancy measures, denoted as $\Delta(M)$ (see Section 2.2 for a formal definition of valid occupancy measures). Nevertheless, since the transition functions are unknown to the learner, $\Delta(M)$ is not known a-priori and must be estimated in an online fashion. On the contrary, the dual player optimizes over the dual variable space, defined as $[0, +\infty)$. Specifically, as a primal optimizer, we employ the state-of-the-art algorithm for unconstrained adversarial MDPs proposed in (Jin et al., 2020). Instead, the dual optimizer consists of a projected gradient descent on the Lagrangian variable $\lambda$.

In Algorithm 2 we provide the pseudocode of PD-DP. In the following, we describe the algorithm in detail.

### 3.1 INITIALIZATION AND LOSS COMPOSITION

The initialization of the primal and dual variables is performed as follows. The occupancy measure is initialized uniformly over the space, while the Lagrangian variable is initialized to 0 (see Line 1).

For every episode $t \in [T]$, the Lagrangian function, namely, the objective of the Lagrangian game, is built given the partial observation received by the environment (see Line 3). We need to emphasize two critical elements of this phase. Firstly, considering the bandit feedback obtained from the environment, the construction of the Lagrangian function is confined to the state-action pairs encountered by the learner, namely, the prices selected by the website. Additionally, it is crucial to note that the Lagrangian function is inverted compared to the formulation in Equation (2). This

reversal is customary in the literature on online adversarial MDPs, as the focus typically shifts to optimizing over losses rather than rewards (see Line 4).

### 3.2 ESTIMATION OF THE UNKNOWN PARAMETERS

In the following, we provide how the unknown loss functions and the transitions of the Markov decision processes are estimated.

#### 3.2.1 PRIMAL LOSS ESTIMATION

The estimation of the loss function is carried out using an optimistic approach. Specifically, in the absence of knowledge about the true transition function $P$, constructing an unbiased estimator $\ell_t(x, a)/q_t(x, a)$ is not feasible. Therefore, we calculate an optimistic occupancy bound. This bound represents an upper limit on the likelihood of reaching the state-action pair $(x, a)$ under the policy $\pi_t$ and within the transition confidence set $\mathcal{P}_i$, which will be detailed in the subsequent section. This is done employing the Comp-UOB function (Line 5) defined as Comp-UOB$(x, a, \pi, \mathcal{P}) := \max_{P_t \in \mathcal{P}} q^{P_t, \pi_t}(x, a)$, and the optimistic biased estimator is computed as:

$$\widehat{\ell}_t(x, a) = \frac{\ell_t(x, a)}{u_t(x, a) + \eta} \mathbb{I}_t(x_{k(x)} = x, a_{k(x)} = a),$$

where $\eta$ is an additional optimistic exploration factor which is initialized as the learning rate, $u_t(x, a)$ is the output of the Comp-UOB function and $\mathbb{I}_t(x_{k(x)} = x, a_{k(x)} = a)$ is the indicator function of the visit for the state-action pair given as argument (Line 6).

#### 3.2.2 EPOCHS AND TRANSITION FUNCTIONS

The method for constructing and utilizing confidence sets on transition probability functions adheres closely to the approach described in (Jin et al., 2020). We include a description of the process here for the sake of completeness. For computational efficiency, the episodes are dynamically segmented into epochs. In particular, the epoch number is updated once a sufficient amount of data regarding the transition function has been collected. Our algorithm keeps counters of visits of each state-action pair $(x, a)$ and each state-action-state triple $(x, a, x')$ in order to estimate the empirical transition function as: $\overline{P}_i(x' \mid x, a) = M_i(x'|x,a)/\max\{1, N_i(x,a)\}$, where $N_i(x, a)$ and $M_i(x' \mid x, a)$ are the initial values of the counters, that is, the total number of visits of pair $(x, a)$ and triple $(x, a, x')$ respectively, before epoch $i$ (see Line 7). Subsequently, the confidence set on the transition functions for epoch $i$ is defined as:

$$\mathcal{P}_i = \left\{ \widehat{P} : \left| \widehat{P}(x'|x, a) - \overline{P}_i(x'|x, a) \right| \leq \epsilon_i(x'|x, a), \quad \forall (x', a, x) \in X \times A \times X \right\}, \quad (3)$$

with $\epsilon_i(x'|x, a)$ defined as $\epsilon_i(x'|x, a) = 2\sqrt{\frac{\overline{P}_i(x'|x,a) \ln\left(\frac{T|X||A|}{\delta}\right)}{\max\{1, N_i(x,a)-1\}}} + \frac{14 \ln\left(\frac{T|X||A|}{\delta}\right)}{3 \max\{1, N_i(x,a)-1\}}$, for some confidence parameter $\delta \in (0, 1)$ (Line 11). In every epoch $i$, the set $\mathcal{P}_i$ serves as a stand-in for the true, yet unknown, transition function $P$.

### 3.3 PER-EPISODE UPDATE

In the following, we provide the update Algorithm 2 performs on the occupancy measure, on the Lagrangian variable and finally, we provide the policy selection procedure.

#### 3.3.1 PRIMAL UPDATE

Given the optimistic biased estimator and the estimated transition functions decision space, it is possible to perform an online mirror descent (OMD, (Orabona, 2019)) update on the primal variable (see Line 12). Precisely, we update the estimated occupancy measure $\widehat{q}_t$ as follows:

$$\widehat{q}_{t+1} = \arg\min_{q \in \Delta(\mathcal{P}_i)} \widehat{\ell}_t^\top q + \frac{1}{\eta} D(q \parallel \widehat{q}_t),$$

where $D(\cdot \parallel \cdot)$ is the unnormalized KL-divergence defined as $D(q_1 \parallel q_2) := \sum_{x,a,x'} q_1(x, a, x') \ln q_1(x,a,x')/q_2(x,a,x') - \sum_{x,a,x'} (q_1(x, a, x') - q_2(x, a, x'))$, and $\Delta(\mathcal{P}_i)$ is

the occupancy measure space where the true transition function is substituted by estimated transitions set $\mathcal{P}_i$.

### 3.3.2 DUAL UPDATE

The dual update is performed by an online gradient descent (OGD (Orabona, 2019)) step over the positive half-space, namely, the Lagrangian function decision space, namely:

$$\lambda_{t+1} \leftarrow \Pi_{\mathbb{R}_{\geq 0}} \left( \lambda_t + \eta \sum_k g_t(x_k, a_k) \widehat{q}_t(x_k, a_k) \right),$$

where $\Pi_{\mathbb{R}_{\geq 0}}(c) := \min_{\lambda \in \mathbb{R}_{\geq 0}} \|c - \lambda\|_2$ is the euclidean projection on $\mathbb{R}_{\geq 0}$ (Line 13).

### 3.3.3 POLICY SELECTION

Differently from standard reasoning in the online learning in MDPs literature, the played policy is not the one trivially induced by $\widehat{q}_t$ (see Equation (1)). Precisely, given the intrinsic partially observable nature of our setting, it is necessary to find a unique policy to play in states $x_1, x_2$. Thus, we decided to perform a convex combination between the policy of the aforementioned state, weighted for an input parameter $\alpha$ which can be estimated beforehand or dynamically during the learning procedure (see Line 14 and Section 4 for additional information).

## 4 SIMULATIONS

In the following section, we focus on the empirical performance of Algorithm 2. We begin by detailing the process for constructing the unknown parameters of the online learning problem, that is, how we extract the online features from the available real-world dataset. Subsequently, we discuss the performance metric used to evaluate our algorithms. Finally, we showcase the results obtained in terms of the aforementioned performance metrics. We refer to the Appendix for the complete simulations of our algorithm.

### 4.1 REAL DATA GENERATION

The parameters of the environment, namely the transition function $P$, rewards $\boldsymbol{r}$, constraints $\boldsymbol{g}$ and conversion rates are estimated using real-world data retrieved from *lastminute.com*, a world's leading online travel agency. Specifically, we employed their user's dataset to infer the distributions of the aforementioned unknown parameters and thus, to build an online simulator for our algorithm.

The dataset, structured as a CSV file, consists of $\sim 4\,000\,000$ rows. Given that our algorithm operates non-contextually, direct utilization of these contextual features is not feasible. However, we can leverage them to cluster the data, enabling the creation of distinct MDPs for each cluster. These MDPs can have unique transition functions, conversion rates, and optimal policies.

In the dataset, each row corresponds to a user search on the website, and each column represents the following key features:

- Unique Search Identification: each time a user searches for a flight (trip) on the site, a unique id_search is generated. This ID distinctly identifies a user's search.
- Associated Data for Each Booking:
  - Status: it indicates the booking's status, with various codes representing different stages, from a trip being clicked to a payment being processed and finalized.
  - Average Flight Cost: the mean cost of the proposed trips.
  - Flight Markup: the markup applied to the flight.
  - Baggage Show Flag: a boolean flag indicating whether baggage options were presented.
- Additional Baggage Information: if baggage was shown, further details are provided, including the baggage price per person, the applied markup, and a boolean flag indicating whether the baggage was sold.
- Additional Search Information:
  - Booking Date: the date when the search was conducted.

- Outbound and Return Dates: dates of departure and return for the searched trip.
    - Roundtrip Flag: a boolean indicating whether the search was for a one-way or round-trip journey.
    - Passenger Quantities: counts of adults, children, and infants.
- Route: the flight route, e.g., ROM-TIA for Rome to Tirana.
- Route Type: specifies the flight type, i.e., short-haul, domestic, or long-haul.

Among these features, the "status" values closely correspond to some of the states in our MDP model. The possible values for "status" are as detailed in Table (1).

Table 1: Status feature values and their correspondences to website progression

| Status Value | Description |
|---|---|
| $nan$ | No flight has been clicked |
| CLICK | A flight has been clicked |
| PAYMENT PAGE | User viewing the final payment page |
| PROCESSING PAYMENT | Payment in process |
| PAYMENT COMPLETED | Payment completed |

The preprocessing steps consist of eliminating rows lacking flight cost information, discarding approximately 7% of the dataset. Next, several features that offer valuable context and can be quantified as numerical values are engineered:

- Lead Time: the time interval (in days) between the booking date and the departure date.
- Length of Stay: the total number of days between the outbound and return dates.
- Number of Passengers: the combined count of adults, children, and infants.

## 4.2 CLUSTERING APPROACH

With the aforementioned extracted features, we proceed to cluster. We scale the numerical features (such as average flight cost and number of passengers) and combine them with the boolean attributes. Employing the K-Means clustering algorithm, we explore $k = 1, \ldots, 10$ clusters and determine the optimal number using the elbow method based on inertia. The chosen number of clusters is $k = 2$, with the first cluster encompassing about 70% of the observations. The clustering segregates flights into two categories: low-to-middle cost flights (cluster 0), primarily short-haul or domestic flights with shorter durations, and high-cost flights (cluster 1), predominantly long-haul flights with longer lead times and durations.

## 4.3 PARAMETER ESTIMATION

For each cluster, we estimate the environment parameters. We determine the discretization level to price the primary product and the ancillary, i.e., we group the "Markup" of the flight and the "Markup" of ancillaries. The number of groups enforces the discretization level. In subsequent simulations, we set the discretization to the simplest possible level, i.e., two possible prices for the primary product and two possible prices for the ancillary. The transition function $P$ is estimated by calculating the mean transition values of *lastminute.com* searches for each price discretization bin, based on the "status" feature detailed in Table (1):

1. In the first layer, from state $x_0$:
    - If the status is PROCESSING PAYMENT or PAYMENT COMPLETED, it implies the primary product was purchased, transitioning the user to state $x_1$.
    - If the status is CLICK or PAYMENT PAGE, it's assumed that the ancillary product was shown, but no purchase happened, transitioning the user to state $x_2$.
    - A "nan" status indicates the user exited, leading to state $x_3$.
2. In the Second Layer:
    - From state $x_1$: the user must proceed to state $x_4$ (final page) as the primary product has been purchased. The rewards are estimated using the conversion rate for actions where ancillary information is available.
    - From state $x_2$: for records with status CLICK or PAYMENT PAGE, the transition depends on whether the final page was viewed (status PAYMENT PAGE leading to state $x_4$) or not (status CLICK leading to State $x_5$).

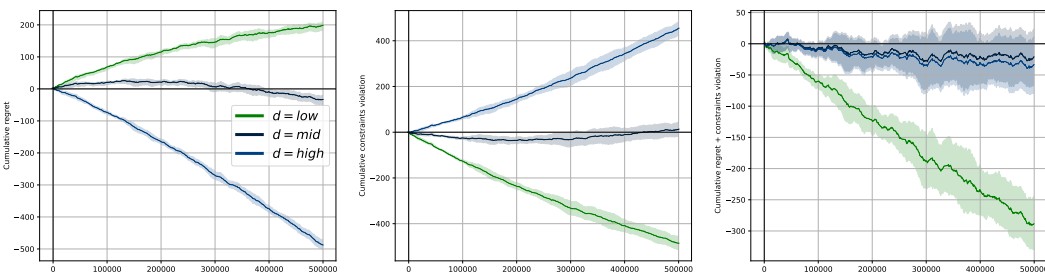

Figure 2: Cumulative Regret (CR), Cumulative Constraints Violation (CCV), and the sum of CR and CCV related to the experiment.

- From state $x_3$: the user invariably proceeds to state $x_5$ as they have exited the website.

The rewards associated with states $x_0$ and $x_1$ correspond to the profits generated from the sale of the primary and the ancillary product, respectively. To simulate these rewards, we employ a Bernoulli distribution $X_{(x,a)}$ for every corresponding state-action pair $(x, a)$. The mean of this distribution is set as the conversion rate for each specific price discretization level. This rate is calculated using the dataset, where it is defined as the ratio of total sales to the total number of site visits for each discrete action (i.e., the discretized "Markup" for primary products and the discretized "Markup" for ancillaries). Additionally, a constant reward is assigned to states $x_1$, $x_2$, and $x_4$. This reward reflects user engagement: in states $x_1$ and $x_2$, it's for engaging with the primary product, and in state $x_4$, it's for interacting with the ancillary product. This fixed reward component acknowledges and incentivizes user engagement within the MDP framework, separate from the direct purchasing actions.

To conclude the section, we describe how the convex combination parameter $\alpha$ has been estimated. $\alpha$ is the weighting parameter of the convex linear combination that forms the output policy. As both policies are discrete distributions over the same action set, their convex linear combination will also be a valid distribution. $\alpha = \alpha(a^*)$ is estimated dynamically as the likelihood of transitioning to $x_1$ for each action $a^* \in A_0$, using the transition function $\bar{P}$. $\alpha_t$ is defined as $\alpha_t := \frac{\overline{P}_i(x_1|x_0,a_t^*)}{\overline{P}_i(x_1|x_0,a_t^*)+\overline{P}_i(x_2|x_0,a_t^*)}$, where $a_t^* \in A_0$ is the action played in the first step of the constrained MDP at the round the convex combination has been performed.

### 4.4 PERFORMANCE MEASURES

In the following, we propose the main performance measures that are employed to evaluate online learning algorithms in constrained settings. Specifically, we define the notion of cumulative regret (CR) as $R_T := \sum_{t=1}^{T} \overline{r}_t^\top q_t^* - \sum_{t=1}^{T} r_t^\top q_t$, where $\overline{r}_t = \mathbb{E}_{r \sim \mathcal{R}_t}[r]$, $q_t^* := \max_{q \in \Delta(M), \overline{g}_t^\top q \leq 0} \overline{r}_t^\top q$ and $\overline{g}_t = \mathbb{E}_{g \sim \mathcal{G}_t}[g]$. As concerns the violations, we define the cumulative constraints violation (CCV) is defined as $V_T := \sum_{t=1}^{T} g_t^\top q_t$. In general, an online algorithm performs properly if the cumulative regret and the cumulative violation are sublinear, namely, $R_T = o(T)$ and $V_T = o(T)$.

### 4.5 EMPIRICAL EVALUATION

To conclude the section, we report a sample experiment conducted in the environment described in Section 4. For the complete experimental evaluation of our algorithm, we refer to the Appendix.

Precisely, in Figure 2, we include the cumulative regret, violation, and the sum of the previous metrics. We consider three settings, which depend on the possibility of satisfying the constraints. Precisely, when the difficulty is high, it is *almost* impossible to satisfy the constraints, while, in the low difficulty case, there exist many occupancies that are feasible. Notably, we observe that the regret grows sublinearly in $T$ in all three settings. Differently, the violation simply depends on the difficulty of the problem. Nevertheless, we observe that both in the low and in the middle difficulty case, the violations are not only sublinear, but they approach $0$ (middle difficulty) or are even negative (low difficulty).

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

APPENDIX

The Appendix is structured as follows:

- In Appendix A, we provide the complete discussion on related works.
- In Appendix B, we provide the results of the experiments simulated in stationary settings.
- In Appendix C, we provide the results of the experiments simulated in non-stationary settings with abrupt changes.
- In Appendix D, we provide the results of the experiments simulated in non-stationary setting with smooth changes.
- In Appendix E, we provide the results of the experiments simulated given a delayed feedback to the algorithm.
- In Appendix F, we provide the results of the experiments simulated given a batch feedback to the algorithm.
- In Appendix G, we provide the comparison between our algorithm's performances and UCB's ones.

## A  RELATED WORKS

Our work is strongly related to the reinforcement learning (RL) literature applied to dynamic pricing and online learning in Markov decision processes theory. Below, we underscore the most cutting-edge contributions in these specified fields of study.

**Reinforcement Learning for Dynamic Pricing**   The use of RL and Markov Decision Processes (MDP) in dynamic pricing contexts is widely recognized. MDPs provide a solid structure for modeling complex dynamic pricing challenges, particularly when pricing multiple items. In a similar vein, RL methodologies, such as Q-learning, are extensively employed to address discrete pricing issues within the MDP framework. This is largely due to their effectiveness in navigating the uncertainties associated with demand and pricing results (e.g., (Kim et al., 2015; Lu et al., 2018; Liu et al., 2021)). Modeling dynamic pricing scenarios with RL frameworks allowed the development of provably optimal algorithms with good empirical performances. The same reasoning holds for simpler single-state dynamic pricing settings, where Multi-Armed Bandits algorithms (see (Ganti et al., 2018; Misra et al., 2019; Trovò et al., 2018)) are used to learn the optimal price, while keeping small the losses that the seller may incur during the learning dynamic.

**Online Learning in Markov Decision Processes**   The literature related to online learning problems (Cesa-Bianchi & Lugosi, 2006) in MDPs is wide (see (Auer et al., 2008; Even-Dar et al., 2009; Neu et al., 2010; Rosenberg & Mansour, 2019b; Jin et al., 2020; Bacchiocchi et al., 2023; Maran et al., 2024)). In such contexts, two primary forms of feedback are commonly examined: the *full-information feedback* model, where the entire loss function is revealed following the learner's decision, and the *bandit feedback* model, in which the learner observes only the incurred loss. Over recent years, online learning in MDPs with constraints has garnered considerable attention. Specifically, the vast majority of prior research in this area focuses on scenarios where constraints are imposed in a stochastic manner (Zheng & Ratliff (2020)). Wei et al. (2018) address adversarial losses and stochastic constraints under the assumption that transition probabilities are known, and the feedback provided is of the full-information type. Bai et al. (2020) introduce the first algorithm able to achieve sub-linear regret in situations where transition probabilities are unknown. This assumes that the rewards are deterministic and the constraints are stochastic, possessing a specific structure. Morover, Efroni et al. (2020) suggest two strategies for navigating the exploration-exploitation trade-off in episodic Constrained Markov Decision Processes (CMDPs). These methods ensure sub-linear regret and constraint violation in scenarios where transition probabilities, rewards, and constraints are both unknown and stochastic, and the feedback received is of the bandit type. Qiu et al. (2020) provide a primal-dual approach based on *optimism in the face of uncertainty*. This research demonstrates the efficacy of this approach in handling episodic CMDPs characterized by adversarial losses and stochastic constraints. It achieves sub-linear regret and constraint violation while utilizing full-information feedback. Stradi et al. (2024b) present a 'best-of-both-worlds' algorithm for CMDPs

operating with full-information feedback. (Stradi et al., 2024a) is the first work to tackle CMDPs with adversarial losses and bandit feedback, while assuming that the constraints are stochastic. Bacchiocchi et al. (2024) study a generalization of stochastic CMDPs with partial observability on the constraints. Finally, Stradi et al. (2024c) study CMDPs with non-stationary rewards and constraints, assuming that the non-stationarity is bounded.

## B  EXPERIMENT 1: STOCHASTIC STATIONARY SETTING

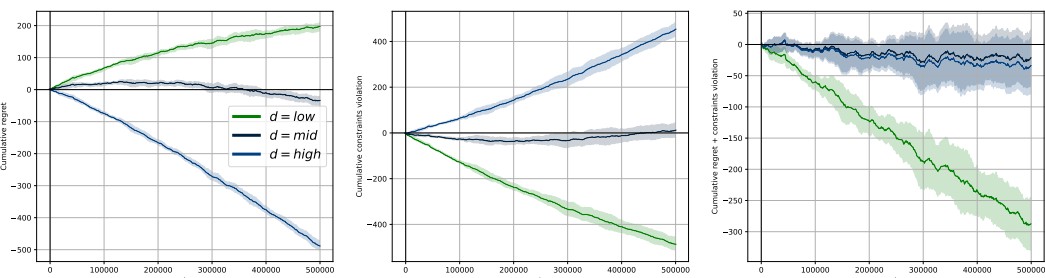

Figure 3: Cumulative Regret (CR), Cumulative Constraints Violation (CCV), and the sum of CR and CCV related to the experiment.

**Rationale**  The first experiment is conceived as a pilot test to study the algorithm's behavior in the simplest configurable settings based on real-world data. Hence, the rewards are stationary, with conversion rates and transition function retrieved from cluster 0 flight data (short-haul, low-cost flights). The update of the policy is carried out at each iteration.

**Parameters**  The confidence delta of the primal algorithm is set to $0.01$. The experiment is run for a temporal horizon of $500\,000$ rounds and iterated for $5$ times to build a confidence interval ($95\%$). We test the algorithm against three different levels of constraints difficulty, i.e., the proportion of sales required to satisfy the constraints. The first one is a low-level difficulty that would still be satisfied on average by any choice of the algorithm. The second is a mid-level difficulty that requires accurate decisions to meet the constraints. The third one is the high-level difficulty that makes constraints satisfaction almost unfeasible.

**Results**  In Figure 3, it is shown that, for low and medium difficulties, the cumulative regret follows expected patterns, with initial rapid increases indicating the algorithm's exploration phase. Over time, the rate of growth in regret flattened, showing the algorithm's adaptation and convergence towards an optimal policy. Notably, lower difficulty levels led to more pronounced initial regret, which stabilized as the simulation progressed. This is due to a counterbalance with the cumulative constraints violation, which is much lower for lower difficulties, yielding an overall more performing curve of the sum of cumulative regret and cumulative constraints violation. As the problem is almost unfeasible for high difficulties, the algorithm focuses on regret minimization, ignoring constraints satisfaction, and the resulting sum curve yields virtually identical performance to the mid-level difficulty.

## C  EXPERIMENT 2: STOCHASTIC NON-STATIONARY SETTING WITH ABRUPT CHANGES

**Rationale**  The second experiment is carried out to evaluate the algorithm's performance under a challenging seasonality setting. The rewards are not stationary and are subject to conversion rates with abrupt changes that are equally distributed throughout the total rounds. Notably, the difficulty of the constraints is much easier after the shifts in conversion rates, as we simulate a high-seasonality setting in which selling a product is more likely. Runs are carried out with parameters estimated on cluster 0 and cluster 1 separately. The update of the policy is carried out at each iteration.

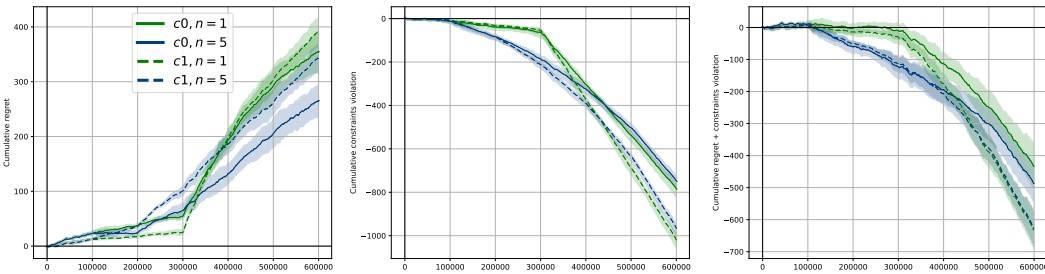

Figure 4: Cumulative Regret (CR), Cumulative Constraints Violation (CCV), and the sum of CR and CCV related to the experiment.

**Parameters** The confidence delta of the primal algorithm is set to $0.01$. The experiment is run for a temporal horizon of $600\,000$ rounds and iterated for $5$ times to build a confidence interval ($95\%$). We test the algorithm with parameters estimated on cluster $0$ or $1$ in two conditions: $1$ bigger abrupt change at half of the simulation rounds, or $5$ smaller equally distributed abrupt changes, denoted by $n = 1$ or $n = 5$ in the plots' legend.

**Results** In both the clusters, the abrupt changes in conversion rates lead the cumulative regret to an upward spike (see Figure 4). Indeed, after the abrupt shifts, the algorithms tend to focus on constraint satisfaction (made easier at the expense of regret minimization). However, notice that the experiments with more frequent abrupt changes (thus, smaller change of conversion rates) appear better than the single (abrupt change) counterpart; indeed, in such cases, the final regret is lower while maintaining a practically identical curve of constraint satisfaction. This effect is particularly visible for cluster $0$ curves. This indicates that the algorithm more easily manages minor shifts.

# D    EXPERIMENT 3: STOCHASTIC NON-STATIONARY SETTING WITH SMOOTH CHANGES

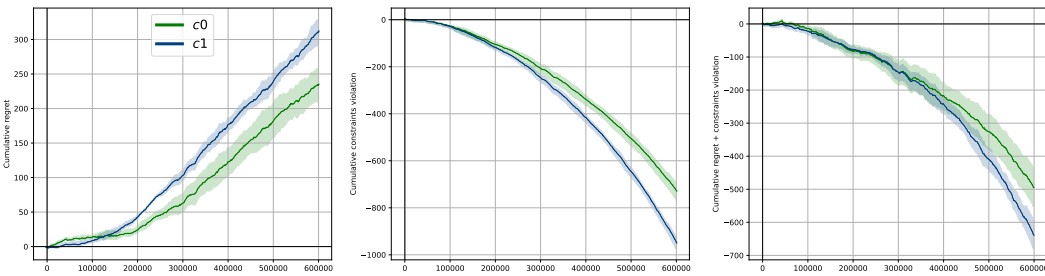

Figure 5: Cumulative Regret (CR), Cumulative Constraints Violation (CCV), and the sum of CR and CCV related to the experiment.

**Rationale** The third experiment is meant to test the algorithm in a non-stationary setting with continuous, small-scale shifts to conversion rates. The experiments' initial and final conversion rates are the same as Experiment C, with constraints being progressively easier to satisfy. Two runs are carried out with parameters estimated on cluster $0$ and cluster $1$ separately. The update of the policy is carried out at each iteration.

**Parameters** The confidence delta of the primal algorithm is set to $0.01$. The experiment is run for a temporal horizon of $600\,000$ rounds and iterated for $5$ times to build a confidence interval ($95\%$). We test the algorithm with parameters estimated either on cluster $0$ or cluster $1$ separately.

**Results** The results are in line with the ones obtained in the second experiment. In both the clusters, the changes in conversion rates lead the cumulative regret to an upward motion see (Figure 5).

Similarly, the algorithms tend to focus on constraint satisfaction, which, in the second phase, turns out to be much easier at the expense of regret minimization. The final regret is still lower than in the previous experiments, confirming that the algorithm more easily manages minor shifts.

## E  EXPERIMENT 4: STOCHASTIC STATIONARY SETTING WITH BATCH DELAYED UPDATE

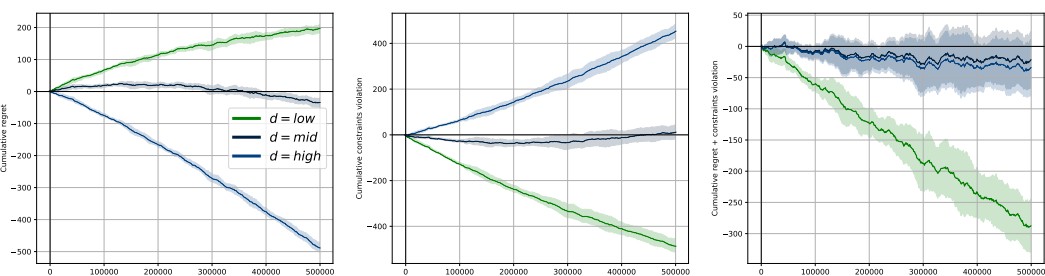

Figure 6: Cumulative Regret (CR), Cumulative Constraints Violation (CCV), and the sum of CR and CCV related to the experiment.

**Rationale**  The fourth experiment tests the algorithm behavior in a production environment, where the policy update is likely carried out in batches. The method of batch update is *delayed*, i.e., the algorithm collects data for n_batch rounds and then pauses and does n_batch updates to the policy. The rewards are stationary, with conversion rates and transition function retrieved from cluster 0 flight data (short-haul, low-cost flights).

**Parameters**  The confidence delta of the primal algorithm is set to $0.01$. The experiment is run for a temporal horizon of $1\,000\,000$ rounds and iterated for $5$ times to build a confidence interval ($95\%$). We test the algorithm against three different levels of constraints difficulty, with the same parameters as Experiment B. The first one is a low-level difficulty that would still be satisfied on average by any choice of the algorithm. The second is a mid-level difficulty that requires accurate decisions to meet the constraints. The third one is the high-level difficulty that makes constraints satisfaction almost unfeasible. The batch size is set to $20$.

**Results**  The results shown in Figure 6 are strongly aligned to the ones in Experiment B in terms of regret and constraint violation value and curve order. Differently from Experiment B, the batch size is increased by a factor of 20 (indeed, the first experiment had a batch size of 1, i.e., no batch); nevertheless, the number of rounds required to achieve the same result increased by only a factor of 2, showing a robust performance to batch updates.

## F  EXPERIMENT 5: STOCHASTIC STATIONARY SETTING WITH BATCH MEAN UPDATE

**Rationale**  The fifth experiment shares similarities with the fourth one, such as the rationale and the environments (see Experiment E); nevertheless, we refer to this kind of batch update as *mean*, i.e., the algorithm collects data for n_batch rounds and performs only $1$ update of the policy using the mean values of rewards and constraints accumulated so far. The algorithm keeps track of the overall count of state-action pairs and state-action-state triples. It updates the counters, epochs, and confidence intervals of the estimated transition function using these counts.

**Parameters**  The confidence delta of the primal algorithm is set to $0.01$. The experiment is run for a temporal horizon of $1\,000\,000$ rounds and iterated for $5$ times to build a confidence interval. We test the algorithm against the same three levels of constraint difficulty as Experiment E. The batch size is set to PAYMENT COMPLETED.

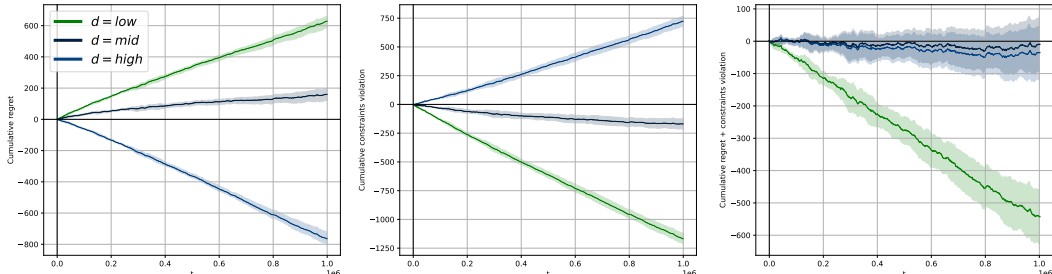

Figure 7: Cumulative Regret (CR), Cumulative Constraints Violation (CCV), and the sum of CR and CCV related to the experiment.

**Results**   In such a setting, the reasoning on the order of the curves and the counterbalance between CR and CCV are the same as in Experiment B and Experiment E; nevertheless, the behavior in terms of regret and sublinearity changes significantly. As shown in Figure 7, when the batch update is performed using empirical means, the algorithms' performance worsens due to the loss of knowledge encountered in the learning dynamic.

# G   EXPERIMENT 6: COMPARISON WITH UCB ALGORITHM

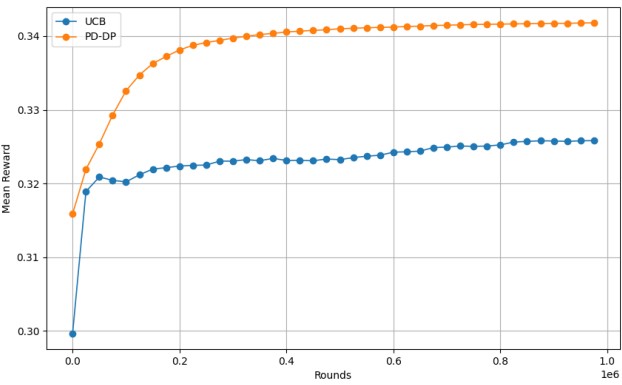

Figure 8: Mean reward comparison (PD-DP against UCB). For the sake of visualization, the mean reward is depicted from around 50, making the performance gap visible, given the small scale of defined rewards.

**Rationale**   In the following experiment, we test the performance of our algorithm against a well-known benchmark, the UCB algorithm (Auer et al., 2002). Unlike previous tests, we are running the algorithm in an unconstrained setting to compare it with UCB. Please notice that, in the experiments, a different UCB procedure is instantiated for every decision state of the Markov decision process. The rewards are stationary, with conversion rates and transition functions retrieved from cluster 0 flight data, specifically from short-haul, low-cost flights.

**Parameters**   The confidence delta of the primal algorithm is set to $0.01$. The experiment is run for a temporal horizon of $1\,000\,000$ rounds. The metric used to compare the algorithms is the mean reward, defined as $\frac{1}{T}\sum_{t=1}^{T} r_t^\top \pi_t$.

**Results**   Figure 8 clearly shows that PD-DP consistently outperforms the UCB algorithm in terms of mean reward. This highlights PD-DP's effectiveness in maximizing rewards under the specified experimental conditions, namely, under correlated multi-product pricing scenarios. Indeed, the UCB instances tend to focus on maximizing their single-step rewards, thus forgetting to optimize the entire

Markov decision process. This behavior prevents the UCB instances from converging to the MDP optimal policy, leading to a smaller mean reward.

