# OpenReview forum: "A Primal-Dual Approach for Dynamic Pricing of Sequentially Displayed Complementary Items under Sale Constraints"
_ICLR.cc/2025/Conference — ICLR 2025 Conference Withdrawn Submission_

### Official Review · Reviewer_pWxD · 2024-10-26

**Soundness:** 2
**Presentation:** 2
**Contribution:** 1
**Rating:** 3
**Confidence:** 3

**Summary:**

The authors study the challenging problem of dynamically pricing complementary items that are sequentially displayed to customers, where a coherent pricing policy is essential. Their setting involves a sales constraint. To address this problem, they formulate it as a Markov Decision Process (MDP) with constraints. Using online learning methods, they propose a primal-dual online optimization algorithm. They demonstrate the effectiveness of their algorithm using semi-real-world datasets.

**Strengths:**

1. They present a framework for pricing complementary items based on a constrained Markov Decision Process (MDP).
2. They develop an algorithm leveraging primal-dual online optimization techniques.

**Weaknesses:**

1. To begin with, the algorithm lacks theoretical analysis. Drawing from prior dynamic pricing literature, it is crucial to establish a theoretical performance guarantee for the proposed algorithm.
2. Additionally, no experiments have been conducted to compare the algorithm against other benchmark methods.

**Questions:**

Please see the weakness.

---

### Official Review · Reviewer_Up1h · 2024-10-31

**Soundness:** 2
**Presentation:** 2
**Contribution:** 2
**Rating:** 3
**Confidence:** 3

**Summary:**

The paper tackles the problem of dynamically pricing complementary items with sales constraints using the online Constrained Markov Decision Process (CMDP) framework. They claim that this is the first work to employ this framework to solve this problem. Their algorithm is an online primal-dual method and they provide an empirical evaluation of their method.

**Strengths:**

The paper is clear and the application (dynamic pricing) is interesting and impactful, furthermore, employing such a modern RL framework (online CMDP) for this problem appears to be novel.

**Weaknesses:**

**Summary of the weaknesses:**

1. There is a lack of novelty in the proposed algorithm: they essentially apply techniques from Stradi et al. (2024a, 2024b).

2. Although the proposed algorithm is clearly based on theory-driven methods, there is no discussion about potential theoretical guarantees.

3. The differences between their works and existing frameworks and algorithms are not clear enough.

4. Regarding the empirical evaluation, the only comparison with a baseline is done in the unconstrained setting.

**Detailed arguments:**

1.  **Similarities with Stradi et al. (2024a, 2024b).** According to my understanding, the proposed algorithm (PD-DP) combines techniques of Stradi et al. (2024a, 2024b). As mentioned in the "Related works" appendix, Stradi et al. (2024b) proposes a method for full-information online CMDPs which cannot be applied apply here because of the bandit feedback. On the other hand, Stradi et al. (2024a) allows to handle bandit feedback online CMDPs but considers hard constraints.

  The proposed algorithm (PD-DP) can be described as follows:

  * It starts by initiliazing the occupancy measure and the lagrangian variables as in Stradi et al. (2024b).

  * Observes the bandit feedback as in Stradi et al. (2024a).

  * Builds the loss as in Stradi et al. (2024b).

  * Computes an upper occupancy bound using Comp-UOB as in Stradi et al. (2024a).

  * Builds an optimistic biased estimator of the loss as in Stradi et al. (2024a).

  * Employs the same of update of the confidence set on the transition functions as in Stradi et al. (2024b) with $\epsilon_t$ taken as in Stradi et al. (2024a).

  * Employs OMD to update the occupancy measure (instead of OGD for Stradi et al. (2024b)).

  * Update the lagrange variable as in Stradi et al. (2024b).

  * Finally, because of a constraint of their specific application the updated policy is modified in such a way that it has the same value on $x_1$ and $x_2$.

  If you agree with this comparison then, this shows, to say the least, that the proposed algorithm (PD-DP) is heavily inspired by the work of Stradi et al. (2024a, 2024b) and in my opinion the paper should emphasize this more.

2. **Discussion on the theoretical guarantees.** Although inspired by works which provide theoretical guarantees, the paper does not take the time to discuss this aspect. I do *not* say that some regret bounds should be provided, but rather that some discussion about these should be present. It could include the opinion of the authors and arguments regarding the possibility of deriving such guarantees.

3. **Differences with existing algorithms and frameworks.** I already mentioned that the differences between existing works and the proposed algorithm should be clearer. This also holds for the framework employed. The differences between existing online CMDP frameworks and your problem are not clear enough.

4. **Baselines for the empirical evaluation.** Although there exist online CMDP algorithms that deals with both the bandit feedback and constraints like that of Stradi et al. (2024a), the only comparison with other methods (which is present in the appendix) considers UCB as a baseline which operates in the unconstrained setting.

**Things to improve the paper that did not impact the score:**
* The mathematical formulation of the model (Section 2.1, end of page 2 and page 3) could be splited into paragraphs to improve readability.

* Adding a conclusion to the paper could add value. If the problem is the space limitation, you could reduce the subsections regarding data generation, clustering and parameter estimation (Subsections 4.1, 4.2, 4.3).

* To show that the online learning algorithm performs properly, you plotted the cumulative regret $R_T$ and the cumulative constraint violation $V_T$ and expect these to be sublinear. An alternative way is to consider the averaged version of these metrics $R_T/T$ and $V_T/T$ and expect these to tend to zero (or less). Maybe you could consider this alternative if you find it improves visualization.

* The difficulty parameter $d$ is defined in the appendix as the proportion of sales required to satisfy the constraints. This definition could be moved to the main body of the paper.

* Since the paper is based on empirical evaluation, the comparison with a baseline could be moved to the main body of the paper instead of the appendix.

**Questions:**

1. Do you agree with my comparison with the work of Stradi et al. (2024a, 2024b) described in the weaknesses part, or did I missed something important?
2. Is it easy to obtain regret bounds on your problem or on the contrary challenging and for which reasons?
3. Is your dynamic pricing problem at hand is precisely an instance of a known online CMDP framework (like those of Stradi et al.) or not, if not so, what are the differences?
4. Regarding the empirical evaluation, why not considering other algorithms that handle both the bandit feedback and constraints like that of Stradi et al. (2024a)?

---

### Official Review · Reviewer_g2GM · 2024-11-03

**Soundness:** 2
**Presentation:** 3
**Contribution:** 1
**Rating:** 3
**Confidence:** 3

**Summary:**

This paper introduces and studies the dynamic pricing of complementary items (such as a flight ticket and associated luggage). The goal is to maximize total revenue in an online manner while ensuring that a minimum number of items are sold. The paper formulates the problem as a constrained Markov Decision Process (CMDP) and designs a primal-dual online learning algorithm to address it. A preliminary experiment is conducted to demonstrate the effectiveness of the proposed algorithm, though it lacks comparisons to other baseline algorithms.

**Strengths:**

This paper introduces an interesting dynamic pricing problem for complementary items. It provides a CMDP formulation for the two-item case and designs an online learning algorithm for this basic formulation.

**Weaknesses:**

Although the topic is interesting, the paper only presents a preliminary set of algorithms and results for a basic model. It lacks more concrete contributions from both theoretical and practical perspectives. In particular:
- from a theoretical perspective, the paper does not provide any formal guarantees on the algorithm's performance. It only proposes an online learning algorithm and evaluates its regret numerically. Additionally, the proposed algorithm is similar to that in (Stradi et al., 2024b), but there is no comparison between the two.

- From a practical perspective, the CMDP formulation only considers one primary item (e.g., flight ticket) and one associated item (e.g., luggage), without modeling the interactions between multiple primary items in dynamic pricing. For instance, in revenue management, dynamic pricing often changes based on the number of remaining items (remaining number of flight tickets), a critical aspect that is overlooked here. Moreover, the claim in Remark 1 that the model and algorithm can be easily extended to more than two items seems difficult, given the complexity in the two-item case.

**Questions:**

- Could you comment on the difficulty in obtaining the regret bound for the online learning algorithm? Could you also provide a comparison between the proposed algorithm and the one in (Stradi et al., 2024b)
- Could you provide a formulation and algorithm, for cases for more than two primary items, and the case for more than two complimentary items?

---

### Note · Authors · 2024-11-19

I have read and agree with the venue's withdrawal policy on behalf of myself and my co-authors.